# Comparison of Health Literacy Assessment Tools among Beijing School-Aged Children

**DOI:** 10.3390/children9081128

**Published:** 2022-07-28

**Authors:** Shuaijun Guo, Xiaoming Yu, Elise Davis, Rebecca Armstrong, Lucio Naccarella

**Affiliations:** 1Centre for Community Child Health, Murdoch Children’s Research Institute, Royal Children’s Hospital, Melbourne, VIC 3052, Australia; 2Melbourne School of Population and Global Health, University of Melbourne, Melbourne, VIC 3053, Australia; elise.davis@summerfoundation.org.au (E.D.); rebecca.rmstrng@gmail.com (R.A.); l.naccarella@unimelb.edu.au (L.N.); 3Department of Paediatrics, University of Melbourne, Melbourne, VIC 3052, Australia; 4Institute of Child and Adolescent Health, School of Public Health, Peking University, Beijing 100191, China

**Keywords:** health literacy measurement, inequities, children, secondary school, cross-sectional

## Abstract

Health literacy is a broad and multidimensional construct, making its measurement and conclusions inconsistent. This study aims to compare the patterning of health literacy using different assessment tools and examine their impact on children’s developmental outcomes. A cross-sectional study was conducted with 650 students in Years 7–9 from four secondary schools in Beijing. Health literacy was measured by the eight-item health literacy assessment tool (HLAT, score range 0–37), the six-item Newest Vital Sign (NVS, score range 0–6), and the 16-item Health Literacy Survey (HLS, score range 0–16). Based on Manganello’s health literacy framework, information on upstream factors (e.g., gender, ethnicity, socioeconomic status) and developmental outcomes (e.g., health-promoting behaviours, health service use, global health status) was collected. Overall, the average scores for health literacy were 26.34 ± 5.89, 3.64 ± 1.64, and 13.72 ± 2.94, respectively, for HLAT, NVS, and HLS. The distribution of health literacy varied by socio-demographics and individual characteristics except for gender, no matter which health literacy assessment tool was used. The magnitude of associations between health literacy, its upstream factors and developmental outcomes was greater when using three-domain instruments (HLAT and HLS) than using single-domain instruments (NVS). The approach to health literacy measurement will influence the conclusion. Using multidimensional assessment tools may better capture a child’s health literacy and contribute to the maximum efficiency and effectiveness of school-based health literacy interventions.

## 1. Introduction

Defined as an individual’s ability to obtain, understand and use health information to maintain and promote good health, health literacy is a key concept underlying everyday health-related decisions [1,2]. In the context of the coronavirus disease of 2019 (COVID-19), health literacy supports one’s decisions on washing hands, maintaining physical distance, and complying with quarantine policies, thus contributing to a more likely successful public health response strategy [3,4,5]. A large body of empirical studies shows that low health literacy is prevalent around the world [6,7] and is associated with a range of adverse health outcomes, including unhealthy health behaviours, ineffective use of health services, and poor health status [8,9,10]. Addressing low health literacy has become a global public health priority, with many countries including it as part of national policies and government initiatives [11,12].

Compared to adults, children have five unique characteristics of health literacy [13], which are called 5 “Ds” in terms of “Disease patterns and health perspectives” (i.e., children are experiencing a unique pattern of health, illness and disability), “Demographic patterns” (i.e., children are vulnerable to health inequalities), “Developmental change” (i.e., children are experiencing a life stage in which physical, emotional, cognitive and social development processes occur), “Dependency” (i.e., children are dependent on their parents, friends and peers when making health decisions) and “Democracy” (i.e., children have their own right to be informed and to participate actively in health decision). Like adult health literacy, child health literacy is a multidimensional concept [14,15], consisting of three domains: functional, interactive, and critical. The functional domain refers to basic skills in reading and understanding health information. The interactive domain denotes advanced skills that allow individuals to extract health information from various forms of communication. The critical domain represents more advanced skills that can be used to critically evaluate health information and take control over health determinants [2]. 

National and international surveys show that low health literacy is prevalent amongst children aged 10–24, ranging from 34% in the USA to 93.7% in China [14,15]. From a human capital perspective [16], early investment in health literacy is the most cost-effective means of improving population health and reducing health inequities. As a child grows, his/her health literacy skills evolve over the life course and empower him/her to better take control over surrounding social determinants of health [17]. In the complex pathways from social determinants of health to health disparities, health literacy is not only a direct and independent determinant of health but also a mediating factor that influences the relationship between other social determinants of health (e.g., socioeconomic status) and health outcomes [18]. There is a clear social gradient in children’s health literacy [19], which in turn contributes to their health and developmental outcomes [20]. Findings from systematic reviews show that low health literacy is a crucial driver of health disparities [20,21,22,23]. However, most of these findings are drawn from the adult population. There remains less known about the relationship between health literacy and health disparities amongst children.

When examining the relationship between health literacy and health disparity, it is important to operationalise health literacy in a particular context among a specific population, given the complex nature of this concept [24]. In the present study, we defined child health literacy as a child’s ability to find, understand, appraise, and use health information in everyday life and applied it to school settings. In addition, based on Pleasant and McCormack’s recommendations for advancing health literacy measurement [25,26], it is necessary to measure health literacy within a theoretical framework to clarify its conceptual meaning and how it is distinct from other related constructs such as self-efficacy and socioeconomic status [25,26]. Currently, more than 20 theoretical frameworks have been proposed in the field of child health literacy [27]. Here, we used Manganello’s health literacy framework [28] as a guide, which has three main modules: (1) upstream factors that may influence health literacy (e.g., gender, ethnicity, socioeconomic status); (2) the construct of health literacy (functional, interactive and critical); and (3) down-stream health outcomes (e.g., health behaviours, health service use) that result from health literacy. This framework was selected because it was informed by the ecological theory [29] and Nutbeam’s health literacy model [2]. The ecological theory highlights that health literacy is not only an individual’s capability to protect and maintain health but also an interactive outcome with the broader environment [30]. Except for socio-demographics, empirical studies have shown that other intrapersonal, interpersonal and environmental factors are also associated with child health literacy [31,32,33]. Examples of these factors are personal self-efficacy [33], health interests [34], social support [31], and school environment [35], which also influence children’s physical, cognitive, and educational outcomes.

Currently, more than 30 health literacy assessment tools have been developed or used amongst children [36,37,38]. Due to lack of consensus, it makes child health literacy measurement not equivalent and results incomparable across contexts. Findings from previous systematic reviews showed that health literacy was mainly measured based on personal health skills such as communicating and appraising health information by international researchers [37]. However, most child health literacy studies in China rely on knowledge-based and behaviour-based measurement tools [39,40,41,42]. It remains unclear how skills-based health literacy instruments perform in Chinese children. In addition, little is known about the comparison of health literacy assessment tools in a single study and how the variation in measurement impacts the quantification of inequities in health outcomes [26,43]. Comparing the strengths and weaknesses of different health literacy instruments in measuring health inequities can inform researchers to better understand the underlying construct of each measure and how they perform unique roles in health disparities research. To our knowledge, few studies have compared and examined different health literacy assessment tools [44,45,46]. Of these studies, the authors mainly targeted the adult population and focused on investigating whether influencing factors of health literacy were consistent or not using different instruments rather than examining their associations with health outcomes. 

To fill the above research gaps, we aimed to investigate the patterning of health literacy by socio-demographic and individual characteristics using multiple assessment tools and quantify the impact of different measurement approaches on developmental outcomes among Chinese children. We proposed two related research questions: (1) Does the association between child health literacy and socio-demographic and individual characteristics differ when using varying health literacy assessment tools? (2) Does the association between child health literacy and developmental outcomes differ when using varying health literacy assessment tools? Findings from the present study will inform study designs regarding which health literacy assessment tools are used in practice, thus contributing to more effective interventions and better health outcomes among school-aged children.

## 2. Materials and Methods

### 2.1. Participants and Settings

The current study is part of a PhD research project [15]. A cross-sectional study was designed to recruit children from four secondary schools in Beijing, China, using convenience and clustering sampling. We used a three-stage cluster sample design according to the Chinese Youth Risk Behaviour Survey in Beijing [47,48]. First, two districts were selected according to their socioeconomic levels, one representing high and the other representing low. Second, two schools in each district were selected based on previous research partnerships and appropriate survey timing (class time, class break time or lunchtime). Third, all students in two whole classes (ranging from 20 to 35 students) from each year level (Year 7, 8, or 9) at each school were invited to complete a self-administered questionnaire. Students with severe cognitive, mental and hearing impairments were not included in the analyses. Passive, opt-out consent was obtained from both parents and students. In total, 661 students were invited to participate in the study, with 11 students declining or excluded for analysis- a response rate of 98.3% (650/661). This sample size was considered acceptable for validation studies of health literacy measurement [49] and association studies using more advanced statistical analyses such as path analyses [50]. Data collection was undertaken in November 2015.

### 2.2. Measures

Based on Manganello’s health literacy framework (35), we designed a questionnaire to measure students’ health literacy, key upstream factors, and developmental outcomes. Further details of each variable are available in Table A1.

#### 2.2.1. Health Literacy Assessment Tools

Three health literacy assessment tools were used to compare whether results are consistent between different measures [26]: the 8-item Health Literacy Assessment Tool (HLAT) [51], the 6-item Newest Vital Sign (NVS) [52], and the 16-item Health Literacy Survey (HLS) [53]. The HLAT and HLS were self-report and three-domain instruments that measured an individual’s ability to access, understand, evaluate, and communicate health information in everyday life [51,53]. In contrast, the NVS was a performance-based and functional measure for reading comprehension and numeracy [52]. These three assessment tools have shown satisfactory or acceptable internal consistency (Cronbach’s alpha = 0.79, 0.54 and 0.82 for HLAT, NVS and HLS, respectively, in our sample) and structural validity among children [54,55,56]. The total score range for each assessment tool was 0–37, 0–6, and 0–16, respectively, with higher scores indicating higher levels of health literacy.

#### 2.2.2. Upstream Factors

Information on a range of upstream factors was collected based on Manganello’s health literacy framework [28]. This included gender (male or female), year level (Year 7, 8 or 9), ethnicity (Han or ethnic minorities), family composition (two parents or lone parent), family socioeconomic status (low, medium or high) measured by the Family Affluence Scale [57], whether had interests in health topics (not interested, not sure, or interested), self-efficacy (continuous, higher scores indicating higher self-efficacy) measured by the 10-item General Self-Efficacy Scale (GSES) [58], social support (continuous, higher scores indicating higher social support) measured by the 12-item Multidimensional Scale of Perceived Social Support (MSPSS) [59], and school environment (continuous, higher scores indicating more positive school environment) measured by the 10-item School Environment Scale (SES) [60].

#### 2.2.3. Developmental Outcomes

*Health-promoting behaviours: *Health-promoting behaviours were measured by five items derived from the global school-based student health survey [61]. They included: the frequency of breakfast eating (“During the past seven days, how often did you have breakfast?”; 1 = 0 days; 8 = 7 days), teeth brushing (“How often do you brush your teeth?”; 1 = never; 5 = more than once a day), cigarette smoking (“On how many occasions have you smoked cigarettes in the last 30 days?”; 1 = never; 7 = 40 times or more), alcohol drinking (“On how many occasions have you drunk alcohol in the last 30 days?”; 1 = never; 7 = 40 times or more) and physical activity (“During the past seven days, on how many days were you physically active for a total of at least 60 minutes per day?”; 1 = 0 days; 8 = 7 days). A total score of health-promoting behaviours is obtained by reversing the scores on ‘cigarette smoking’ and ‘alcohol drinking’ and then summing scores across all five items. The total score ranged from 5–35, with higher scores indicating more health-promoting behaviours.

*Body mass index: *Self-reported height and weight were obtained using the questions “How tall do you think you are?” and “How much do you think you weigh?” These two self-reported items are commonly used among children [62]. Body mass index (BMI) was calculated using the following formula: BMI = weight (kg)/height (m)^2^. Continuous BMI values were used for next-step data analysis, with higher scores indicating a high probability of overweight and obesity.

*Global health status: *Global health status was assessed using a widely-used general self-report health question (‘In general, would you say your health is?’ 1 = poor, 5 = excellent) [63]. This single question has demonstrated strong predictive validity with objective indicators of health and mortality [64]. Global health status scores ranged from 1 to 5, with higher scores indicating better health status.

*Health-related quality of life: *Health-related quality of life (HRQoL) was measured by the KIDSCREEN-10 [65], which assesses the health-related quality of life of healthy and chronically ill children aged 8 to 18. Students answered each item on a 5-point Likert scale (1 = not at all/never, 5 = extremely/always). The KIDSCREEN-10 has high internal consistency (Cronbach’s α = 0.79) and strong structural validity (χ^2^/*df* = 2.877, CFI = 0.959, RMSEA = 0.055) in our sample. The KIDSCREEN-10 score is obtained by reversing the scores on two items and then summing scores across all ten items. The total score ranged from 10 to 50, with higher scores indicating higher levels of HRQoL.

*Health service use: *Health service use was assessed using a single item that asked students’ frequency of patient-provider communication over the last 12 months (‘how many times have you raised a question during your doctor’s appointment in the last 12 months?’; 1 = 0 times, 2 = 1–2 times, 3 = 3–5 times, 4 = 6 times or more). This single question has shown satisfactory known-group validity among children [33]. Patient-provider communication scores ranged from 1 to 4, with higher scores indicating more frequency of communication.

*Academic performance: *Academic performance was self-reported by students using a single item that asked them “think of your marks at school, if putting them all together, where were your marks like last year?” (1 = very poor, 2 = poor, 3 = average, 4 = good, 5 = very good). This single item was derived from the Chinese Youth Risk Behaviour Surveillance Survey and has shown strong predictive validity with children’s health outcomes [66]. Academic performance scores ranged from 1 to 5, with higher scores indicating higher academic achievement.

### 2.3. Statistical Analysis

Participant characteristics were summarised using descriptive statistics. The distribution of health literacy was examined by socio-demographics and by each health literacy assessment tool. Correlation analysis (Pearson and Spearman correlation analysis) were conducted to examine the associations between health literacy, its upstream factors and developmental outcomes. Next, a series of multivariable linear regression models were conducted to examine associations between health literacy, its upstream factors and developmental outcomes. We obtained unadjusted estimates and those adjusting for covariates. While the *p*-value is the most commonly used inferential statistic, it is often misunderstood and misinterpreted in the literature [67]. In the main text, we avoided using terms such as “statistically significant” and “non-significant.” Instead, we reported all statistical results using the estimate and its 95% confidence interval. When interpreting the findings, we avoided dichotomising the results but reported the magnitude of associations.

### 2.4. Missing Data

The analytic sample consisted of those who had at least one developmental outcome (*n* = 650). The proportion of students with complete data was 85.2% in our sample. The percentage of missing data ranged from 0.2% to 8.2% across all study variables (Table A2). Multiple imputation by chained equations was conducted to handle missing values for all study variables under the missing at random assumption [68]. We imputed continuous variables using linear regression models and binary variables using logistic regression models. The imputation model included all study variables. Twenty imputed data sets were created, with pooled results combined using Rubin’s rules [69]. Descriptive results are shown using observed data, and association results are shown using multiply imputed data. All analyses were conducted using Stata 17.0 [70].

## 3. Results

### 3.1. Sample Characteristics

In our sample, the mean age of participants was 13.42 (range: 11–17 years), with a standard deviation of 1.01. Students’ gender and year level were evenly distributed. Most students were from Han families (94.9%) and two-parent families (88.1%). Almost one quarter (27.7%) of students came from low-affluence families and one quarter (26.0%) from high-affluence families. The distributions of health literacy, self-efficacy, social support, school environment, and each outcome in the overall sample are shown in Table 1.

### 3.2. Distribution of Health Literacy by Socio-Demographics and Individual Characteristics

The observed distribution of health literacy was examined according to each socio-demographic variable (Table A3). Standardised estimates and 95% confidence intervals are shown by each health literacy assessment tool (Figure 1). Overall, there was no clear difference in health literacy by gender, no matter which health literacy assessment tool was used. In terms of year level, students from Year 8 on average tended to have higher levels of health literacy than those from Year 7 and Year 9. As for ethnicity, children from ethnic minority families and lone-parent families were more likely to have lower health literacy scores than their peers from Han and two-parent families when HLAT and HLS were used to measure health literacy, but the confidence intervals were wide. No matter which approach was used to measure health literacy, children from high-affluence families and those who were interested in health topics had higher health literacy scores than their counterparts from low-affluence families and those who did not have health interests.

### 3.3. Association between Health Literacy and Its Upstream Factors

Correlation analysis showed that health literacy scores measured by HLAT, NVS and HLS were positively correlated with health interest, self-efficacy, social support and school environment (r = 0.10–0.44, *p* < 0.05) (Table 2). Overall, the association between health literacy and its upstream factors varied by each health literacy assessment tool and each upstream factor, after adjusting for all potential confounders (Figure 2, Figure 3 and Figure 4). There was no clear difference in health literacy scores by gender. A small association between health literacy and year level was observed when using NVS and HLS, with children from higher year levels having higher health literacy. Children from ethnic minority families had lower levels of health literacy when using HLS. Hypothesised differences in health literacy were observed between children from long-parent and two-parent families when using HLAT and NVS, but confidence intervals were wide. The association between health literacy and socioeconomic status was in the expected direction when using HLS. There was consistent evidence for the expected association between health literacy and health interest, self-efficacy, social support, and school environment, no matter which health literacy assessment tool was used. Further details are available in Table A4.

### 3.4. Association between Health Literacy and Developmental Outcomes

Correlation analysis showed that students’ health literacy was positively correlated with most developmental outcomes except BMI (r = 0.11–0.35, *p* < 0.05) (Table 2). Figure 5 shows the association between health literacy and each developmental outcome by each health literacy assessment tool. In the unadjusted models, children with higher levels of health literacy were more likely to have more health-promoting behaviours, lower BMI, more frequent patient-provider communication, better health status, higher ratings of HRQoL, and higher levels of academic performance. The magnitude of associations varied by the approach to measuring health literacy. When adjusting for all confounders, we found that the effect sizes of all associations were attenuated. Overall, small associations were observed between health literacy and health-promoting behaviours, patient-provider communication, global health status, and academic performance when using HLAT. While associations between health literacy and developmental outcomes (i.e., health-promoting behaviours, BMI, patient-provider communication, HRQoL, and academic performance) were in the expected direction when using NVS, the confidence intervals were wide. Hypothesised differences in health-promoting behaviours, BMI, HRQoL, and academic performance were observed when using HLS. Further details of these results are available in Table A5.

## 4. Discussion

### 4.1. Summary of Key Findings

The present study used multiple health literacy assessment tools to investigate the patterning of health literacy by socio-demographic and individual characteristics and quantify the impact of different measurement approaches on developmental outcomes among Chinese secondary school students. Confirmed with previous findings [14,19,71,72,73,74], we found that health literacy was associated with a range of upstream factors and developmental outcomes in children, no matter which approach was used to measure health literacy. Different health literacy assessment tools provide varying specificity for quantifying population socio-demographics, individual characteristics, and developmental outcomes. Overall, inequities in health literacy and developmental outcomes appeared to be more prominent when using comprehensive assessment tools, such as HLAT and HLS.

Corresponding to our first research question, we found that variation in health literacy measurement resulted in differing associations with sociodemographics and individual characteristics. With regard to gender, the distribution of health literacy was similar between boys and girls. While differences in health literacy have been observed by gender in previous studies [40,46,75], the authors used knowledge-based, behaviour-based or functional domain tools to measure health literacy. In another two recent studies using skills-based health literacy instruments, Fretian et al. [19] and Paakkari et al. [72] did not find differences in health literacy by gender either. Possible reasons for our findings are that differential health literacy assessment tools may influence the conclusion, and gender-related health literacy differences may emerge later. In terms of year level, we found children from Year 8 had the highest health literacy scores than their peers from Year 7 and Year 9, no matter which health literacy instrument was used. While previous studies showed that the levels of health literacy increased as a child grew [72,76], children in Year 9 with the oldest age did not have the highest health literacy scores. One potential reason is that, due to high levels of academic stress in Year 9 [77], children are likely to pay more attention to academic performance than health literacy. When examining inequities in health literacy by ethnicity, we found that children from ethnic minority families showed lower levels of health literacy scores on HLS than those from Han families, but not on HLAT and NVS. This finding is similar to previous studies that showed differences in the association between race/ethnicity and multiple health literacy assessment tools among adults [44,46]. The underlying explanation might be that HLS is a more comprehensive assessment tool than HLAT and NVS, which measures health literacy not only within three domains (functional, interactive and critical) but also across three dimensions (health care, disease prevention, and health promotion) [53]. Therefore, it may better capture all aspects of a child’s health literacy. This is also the case when examining our sample’s socioeconomic inequities in health literacy. Differences in health literacy scores on HLAT and HLS were observed by family composition but not on NVS. The NVS is a performance-based measure focusing on literacy and numeracy [56], thus probably not comprehensively reflecting a child’s health literacy. Aligning with findings from previous studies [19,76], we also found health literacy inequities exist for other individual characteristics such as health interest, self-efficacy and social support. Children with higher levels of health interest, self-efficacy, and social support and those who felt a more positive school environment had higher health literacy scores on HLAT and HLS. Overall, consistent with the policy statement and empirical findings [12,21,22,78] among adults, health literacy was found as a critical driver of health equity in children. All these findings suggest that HLAT and HLS may better capture a child’s health literacy than NVS. In response to addressing health literacy inequities [18,79,80], intervening on upstream determinants of health is worthwhile through strategies, such as cash transfers to low-income families and improving personal self-efficacy skills in school-aged children.

With regards to our second research question, this study extends current understanding of the association between child health literacy and a range of developmental outcomes using multiple health literacy assessment tools [44,45,46]. Consistent with previous findings [14,71,72,74,81,82], we found that high health literacy in children was associated with a series of positive outcomes, including more health-promoting behaviours, lower BMI, frequent patient-provider communication, better health status, higher HRQoL, and higher academic performance. However, the magnitude of associations varied by each health literacy assessment tool. For example, health literacy scores on HLAT were more related to health-promoting behaviours, patient-provider communication, global health status, and academic performance. In contrast, health literacy scores on NVS had larger effects on BMI, health-related quality of life and academic performance. This again might be due to the nature of each health literacy assessment tool. NVS focuses on functional health literacy measurement, which is more positively correlated with children’s academic performance [56] and HRQoL [83] but negatively with BMI [84]. Findings from the present study are helpful for researchers to ascertain which health literacy measures to select when evaluating an intervention’s effectiveness by a specific outcome indicator. Meanwhile, findings will also inform intervention strategies and contents for school-based interventions that aim to improve health literacy and other health outcomes. If a school-based intervention program focuses on health behaviours as outcome indicators, health literacy interventions need to focus on not only communication of knowledge, but also teaching children personal skills and empowerment to protect and maintain good health [85].

### 4.2. Strengths and Limitations

Compared with previous similar studies [51,52,53], we used both objective and subjective health literacy assessment tools and focused on more than functional assessment. In addition, we used Manganello’s health literacy framework as a conceptual guide to enhancing the rigour, transparency, and clarity of the current study. However, there are several limitations. First, this study only used cross-sectional data to examine inequities in health literacy and developmental outcomes simultaneously. Longitudinal cohort studies are needed in future to replicate our findings. Second, convenience sampling may limit the generalizability of our findings. We recruited students from four secondary schools in a metropolitan city where the ability of children to access good education might be much higher than the general population. Future studies are recommended to recruit children from a broader range of socio-demographic backgrounds. Third, self-report bias may exist for subjective health literacy and other measurement scales such as BMI and health status. However, we used well-established and valid instruments in the present study to minimise the extent of such bias. Despite these limitations, the present study adds significant value to examining health literacy inequities and informing opportunities to reduce these inequities. 

### 4.3. Implications and Future Directions

According to Nutbeam’s health promotion model [2], improving health literacy is an integral part of improving population health and reducing health inequities. However, low health literacy should not be treated as an individual deficit. As shown in our study, a child’s health literacy is influenced by not only socio-demographic and individual characteristics but also social connections and contexts. These insights into health literacy inequities align with the whole-school approach [86,87], highlighting the necessity of considering multi-level interventions and collaborations between families, schools and communities. For example, the HealthLit4Kids program utilises a holistic approach to improving health literacy [88], with a focus on increasing equity in health outcomes for children with varying health literacy needs. 

Empirical research suggests upstream factors of health literacy may interact with each other and play different roles in child health literacy [89,90]. For example, family socioeconomic status has been found as a moderator in the relationship between health literacy and health outcomes such as health behaviours and health service use in the Irish population [90]. Future research may consider examining and comparing the magnitude of associations between health literacy and developmental outcomes within different levels of socio-demographic and individual characteristics (e.g., gender, ethnicity, social support) in children. While improving health literacy may reduce inequities in health and developmental outcomes amongst all children, it may have a more marked impact among some subgroups of children (e.g., low socioeconomic status). Interventions that target disadvantaged children, such as those from low socioeconomic status backgrounds, have shown promising health outcomes, including improved health literacy and health outcomes [91]. These findings will further inform the design of school-based intervention programs and contribute to their efficiency and effectiveness.

## 5. Conclusions

We found that inequities in health literacy and developmental outcomes already exist in late childhood. Different approaches to health literacy measurement may result in different quantification of inequities in health literacy and developmental outcomes among school-aged children. Using multidimensional and comprehensive assessment tools may better capture a child’s health literacy and contribute to the maximum efficiency and effectiveness of school-based health literacy interventions. When designing health literacy interventions that aim to reduce health inequities, researchers need to consider the specificity of health literacy assessment tools and outcome indicators.

## Figures and Tables

**Figure 1 children-09-01128-f001:**
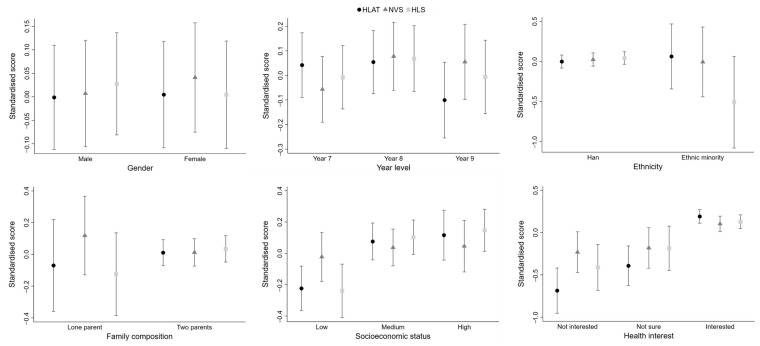
Distribution of health literacy by socio-demographic and individual characteristics.

**Figure 2 children-09-01128-f002:**
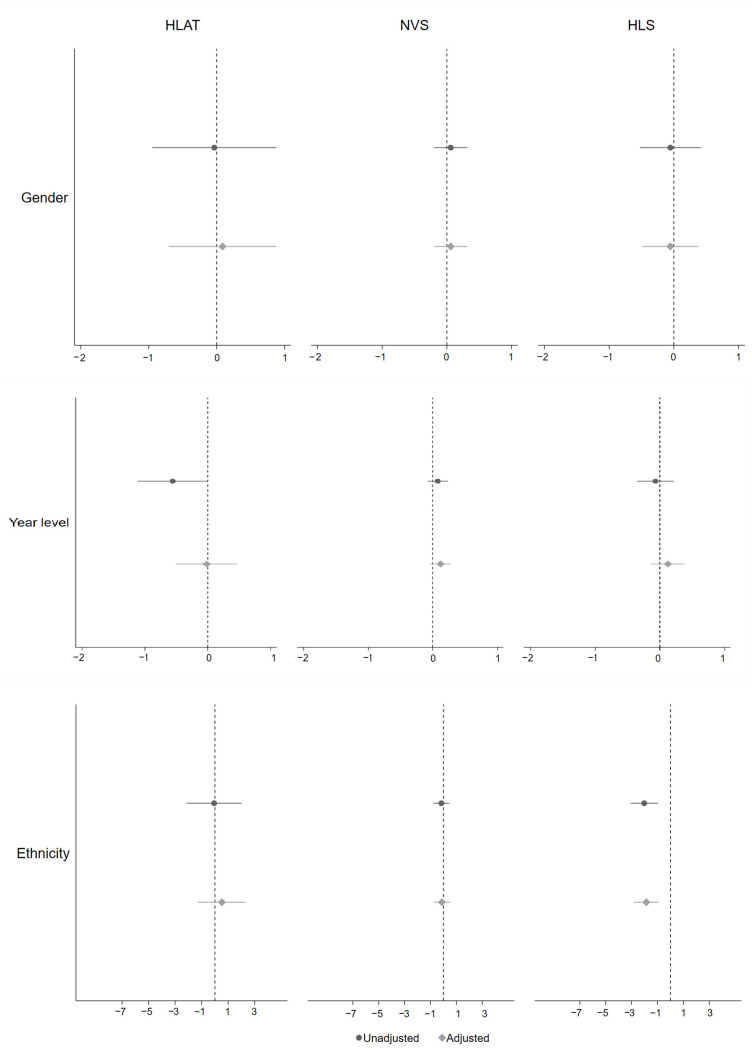
Association between health literacy and gender, year level, and ethnicity. Adjusted confounders were all upstream variables.

**Figure 3 children-09-01128-f003:**
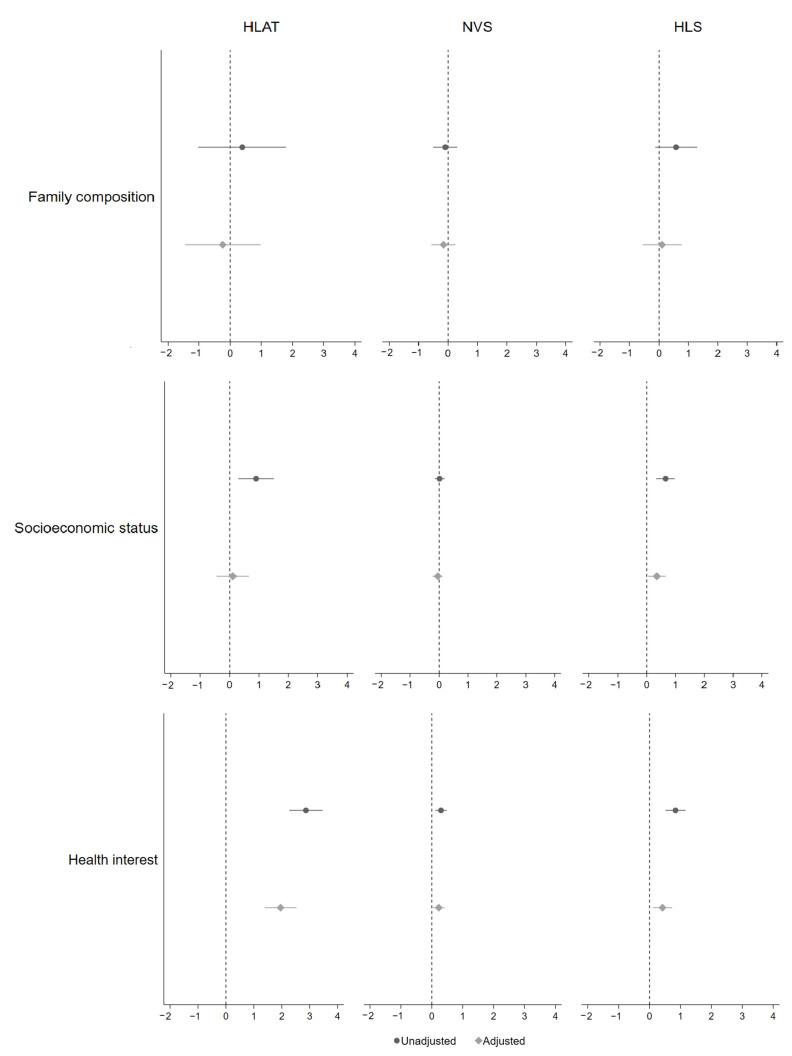
Association between health literacy and family composition, socioeconomic status, and health interest. Adjusted confounders were all upstream variables.

**Figure 4 children-09-01128-f004:**
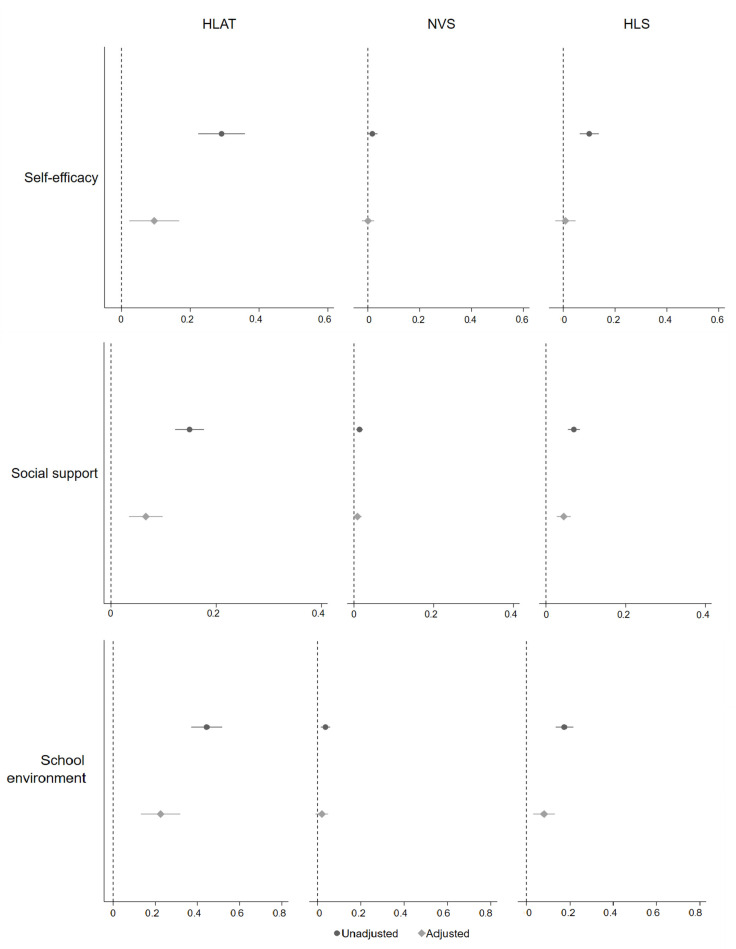
Association between health literacy and self-efficacy, social support, and school environment. Adjusted confounders were all upstream variables.

**Figure 5 children-09-01128-f005:**
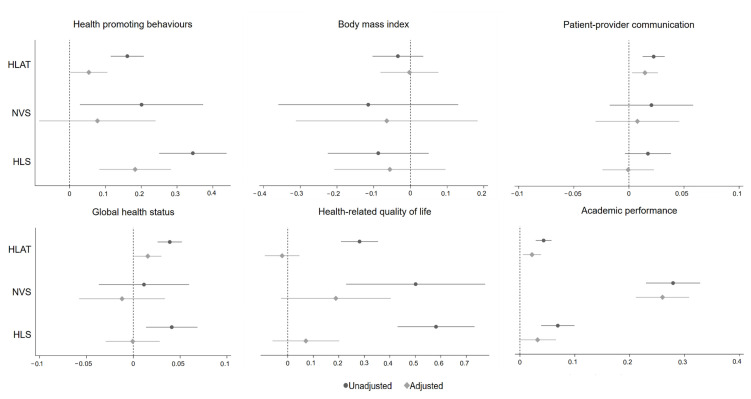
Association between health literacy and developmental outcomes. All estimates were adjusted for gender, year level, ethnicity, family composition, family socioeconomic status, health interest, self-efficacy, social support, and school environment.

**Table 1 children-09-01128-t001:** Sample characteristics and descriptive statistics of key variables (*n* = 650).

Variable	Frequency (%)/Mean (±SD)
*Sociodemographics and individual characteristics*	
Gender	
Male	357 (54.9)
Female	293 (45.1)
Year level	
Year 7	232 (35.7)
Year 8	215 (33.1)
Year 9	203 (31.2)
Ethnicity	
Han	617 (94.9)
Ethnic minority	33 (5.1)
Family composition	
Lone parent	77 (11.9)
Two parents	572 (88.1)
Socioeconomic status	
Low	180 (27.7)
Medium	301 (46.3)
High	169 (26.0)
Health interest	
Not interested	88 (13.5)
Not sure	85 (13.1)
Interested	477 (73.4)
Self-efficacy	26.85 (6.37)
Social support	62.79 (15.26)
School environment	30.48 (5.59)
** *Health literacy* **	
HLAT	26.34 (5.89)
NVS	3.64 (1.64)
HLS	13.72 (2.94)
** *Developmental outcomes* **	
Health-promoting behaviours	28.04 (3.65)
Body mass index	21.21 (5.02)
Patient-provider communication	
0 times	332 (53.2)
1–2 times	221 (35.4)
3–5 times	51 (8.2)
6 times or more	20 (3.2)
Global health status	
Poor	9 (1.4)
Fair	215 (33.1)
Good	227 (34.9)
Very good	125 (19.2)
Excellent	74 (11.4)
Health-related quality of life	37.49 (5.78)
Academic performance	
Very poor	69 (10.6)
Poor	139 (21.5)
Average	197 (30.4)
Good	186 (28.7)
Very good	57 (8.8)

HLAT, Health Literacy Assessment Tool; HLS, Health Literacy Survey; NVS, Newest Vital Sign; SD, Standard Deviation.

**Table 2 children-09-01128-t002:** Correlation between health literacy, its upstream factors and outcomes.

	Gender	YL	Ethnicity	FC	SES	HI	SEF	SS	SCE	HLAT	NVS	HLS	HPB	BMI	PC	GHS	HRQOL	AP
Gender	1.00																	
YL	0.00	1.00																
Ethnicity	−0.01	0.03	1.00															
FC	0.03	0.00	−0.08	1.00														
SES	0.02	−0.01	0.06	0.06	1.00													
HI	0.03	−0.08	−0.04	0.00	0.04	1.00												
SEF	−0.11 *	−0.11 *	−0.01	0.05	0.17 *	0.17 *	1.00											
SS	0.02	−0.06	0.03	0.10 *	0.18 *	0.25 *	0.41 *	1.00										
SCE	0.03	−0.16 *	−0.03	0.03	0.13 *	0.22 *	0.45 *	0.55 *	1.00									
HLAT	−0.00	−0.04	0.03	0.00	0.15 *	0.29 *	0.38 *	0.44 *	0.42 *	1.00								
NVS	0.01	0.03	0.02	−0.02	0.02	0.11 *	0.10 *	0.17 *	0.11 *	0.20 *	1.00							
HLS	−0.03	0.01	−0.07	0.06	0.11 *	0.16 *	0.25 *	0.37 *	0.32 *	0.43 *	0.14 *	1.00						
HPB	−0.07	−0.08	−0.04	0.05	0.12 *	0.17 *	0.30 *	0.28 *	0.32 *	0.29 *	0.07	0.32 *	1.00					
BMI	−0.10 *	0.07	0.01	0.05	−0.03	−0.06	−0.04	−0.03	−0.05	−0.05	−0.05	−0.03	−0.09 *	1.00				
PC	0.05	0.01	0.02	0.04	0.12 *	0.16 *	0.12 *	0.14 *	0.09 *	0.15 *	0.06	0.11 *	0.06	−0.03	1.00			
GHS	−0.14 *	−0.10 *	−0.04	0.11 *	0.10 *	0.14 *	0.25 *	0.23 *	0.19 *	0.24 *	0.02	0.23 *	0.18 *	−0.12 *	−0.05	1.00		
HRQOL	−0.12 *	−0.15 *	−0.04	0.05	0.18 *	0.20 *	0.35 *	0.59 *	0.48 *	0.35 *	0.14 *	0.34 *	0.32 *	−0.06	0.06	0.34 *	1.00	
AP	0.08	−0.01	0.04	−0.01	0.16 *	0.10 *	0.20 *	0.24 *	0.21 *	0.22 *	0.39 *	0.18 *	0.17 *	−0.09 *	0.12 *	0.03	0.22 *	1.00

YL, Year Level; FC, Family Composition; SES, Socioeconomic Status; HI, Health Interest; SEF, Self-efficacy; SS, Social Support; SCE, School Environment; HLAT, Health Literacy Assessment Tool; NVS, Newest Vital Sign; HLS, Health Literacy Survey; HPB, Health-promoting Behaviours; BMI, Body Mass Index; PC, Patient-provider Communication; GHS, Global Health Status; HQROL, Health-related Quality of Life; AP, Academic Performance. * *p* < 0.05.

## Data Availability

The datasets used and/or analysed in the current study are available from the corresponding author on reasonable request.

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
