# Peer review of "Comparison of Health Literacy Assessment Tools among Beijing School-Aged Children"

_children, 2022, doi:10.3390/children9081128_

Round 1

Reviewer 1 Report

Dear Authors,

I have carefully read the interesting manuscript titled “Comparison of Health Literacy Assessment Tools among Beijing School-aged Children”.

I have some suggestions that maybe the authors would like to follow to strengthen the clarity of their work.

-        Please, describe in more detail Manganello’s Health Literacy Framework in the introduction. In this direction, please, report further reference on previous studies underpinning your choice of including specific socio-demographic, socio-economic and other dimensions (e.g., social support) in this study.

-        Please, include research hypotheses and/or research questions

Indeed, while reading, it seems there are several (maybe too many) dimensions and different study aims (e.g., comparing three tools assessing Health Literacy and exploring the role of different predictors of Health Literacy). Moreover, the issue of inequities seems of particular interest, but it seems lost in some points (mainly the introduction and the discussion should consider strengthen this issue). Including explicit hypotheses/research questions would help the reader to understand the study rationale and to follow the paper (I suggest editing method section, results, and discussion according to hypotheses and/or research questions.

-        Table 1 could be split into two. This would help its readability and relevant information could be better appraised.

I wish the authors would follow my suggestions!

Sincerely

Author Response

Thank you for your time and suggestions. We have updated the manuscript according to each comment. Please see the Response form for further details.

Reviewer 2 Report

This manuscript could be an important reference for future studies. However, is still needed to improve the quality of this paper. Please revise the manuscript to address the expressed concerns. After thorough review, I am recommending some revisions. In this regard, kindly address the following comments and suggestions to further improve your manuscript

a.       It was better if you wrote some of main finding as quantitative or mean ±SD within the abstract. The result section in the abstract is poor and immature!!

b.      The introduction needs some revision . You could summarize this section a bit more for readers. Write about the problems, the novelty of your study, and your study goals within the introduction. In this section, you can use the following articles:

1- “ School-Based Health Literacy Educational Interventions in Adolescents: A Systematic Review

2- “Assessment of health literacy with the Newest Vital Sign and its correlation with body mass index in female adolescent students

c.       The materials & methods section is relatively immature. You could expand it a bit more clearly for readers. For example, write about the sample size. How did you calculate sample size for this survey? Where have you collected samples? Write the year and the name of place in which you had done this survey. Furthermore, write about all applied exclusion and inclusion criteria a bit more clearly by which you selected samples for this survey.

d.      What was your sample size formula? What is your expected power? please mention in main text

e.        Discuss more about your sampling strategy? The structure of your sampling is so vague and understandable. Did you have sampling frame? how did you access to this frame

f.         What are the data extract’s center characteristics? is it governmental or private, is it referral or not referral and so on, discuss more about it

g.       Describe the validity and reliability of the measurement tool

h.      Mention the possible score (range) for each scales and meaning of it so easier to readers interpret the results.

i.        - In the discussion, you did not include related previous studies in relation to the findings of the current study. Please search and cite related studies and include them in your discussion.

j.        You could increase the number of more recently studies in the reference section. You should have comprehensive and reliable comparisons between your findings with the other previous studies. Furthermore, write about the limitations of your survey. Are there any limitations for this study? If yes, please mention all limitations of current study within the discussion section, too

k.       Please mention the weak and strong points of your study

l.        There are some spelling and grammatical errors in the text. Please correct them

Author Response

(The authors gave the same response as above.)

Round 2

Reviewer 1 Report

Dear authors, thank you for your work!

Sincerely

Reviewer 2 Report

Accept